# Betanin, a Natural Food Additive: Stability, Bioavailability, Antioxidant and Preservative Ability Assessments

**DOI:** 10.3390/molecules24030458

**Published:** 2019-01-28

**Authors:** Davi Vieira Teixeira da Silva, Diego dos Santos Baião, Fabrício de Oliveira Silva, Genilton Alves, Daniel Perrone, Eduardo Mere Del Aguila, Vania M. Flosi Paschoalin

**Affiliations:** Instituto de Química, Universidade Federal do Rio de Janeiro, Av. Athos da Silveira Ramos 149, 21941-909 Rio de Janeiro, Brazil; daviufrj@outlook.com (D.V.T.d.S.); diegobaiao20@hotmail.com (D.d.S.B.); silvafo@live.com (F.d.O.S.); geniltonalves@gmail.com (G.A.); perrone@iq.ufrj.br (D.P.); emda@iq.ufrj.br (E.M.D.A.)

**Keywords:** beetroot, betalains, semi-preparative RP-HPLC, in vitro human gastrointestinal digestion, ex vivo colon fermentation, antioxidant ability, malonildialdehyde

## Abstract

Betanin is the only betalain approved for use in food and pharmaceutical products as a natural red colorant. However, the antioxidant power and health-promoting properties of this pigment have been disregarded, perhaps due to the difficulty in obtaining a stable chemical compound, which impairs its absorption and metabolism evaluation. Herein, betanin was purified by semi-preparative HPLC-LC/MS and identified by LC-ESI(+)-MS/MS as the pseudomolecular ion *m*/*z* 551.16. Betanin showed significant stability up to −30 °C and mild stability at chilling temperature. The stability and antioxidant ability of this compound were assessed during a human digestion simulation and ex vivo colon fermentation. Half of the betanin amount was recovered in the small intestine digestive fluid and no traces were found after colon fermentation. Betanin high antioxidant ability was retained even after simulated small intestine digestion. Betanin, besides displaying an inherent colorant capacity, was equally effective as a natural antioxidant displaying peroxy-radical scavenger ability in pork meat. Betanin should be considered a multi-functional molecule able to confer an attractive color to frozen or refrigerated foods, but with the capacity to avoid lipid oxidation, thereby preserving food quality. Long-term supplementation by beetroot, a rich source of betanin, should be stimulated to protect organisms against oxidative stress.

## 1. Introduction

Beetroot (*Beta vulgaris L*.) is a vegetable presenting significant scientific interest, mainly because it is a rich source of nitrate (NO_3_^−^), a compound with beneficial cardiovascular health effects, through the endogen production of nitric oxide (NO) [1]. Moreover, beetroots are the main source of betalains, a heterocyclic compound and water-soluble nitrogen pigment, which can be subdivided into two classes according to their chemical structure: betacyanins, such as betanin, prebetanin, isobetanin and neobetanin, responsible for red-violet coloring, and betaxanthins, responsible for orange-yellow coloring, comprising vulgaxanthin I and II and indicaxanthin [2,3]. Betalains are present in the tuberous part of beetroots, conferring its red-purple coloration.

Betanin (betanidin 5-*O*-β-d-glucoside) is the most abundant betacyanin and the only one approved for use as a natural colorant in food products, cosmetics and pharmaceuticals, under code EEC No. E 162 by the European Union and under Section 73.40 in Title 21 of the Code of Federal Regulations (CFR) stipulated by the Food and Drug Administration (FDA) in the United States [4,5,6] (Appendix A).

In the food industry, synthetic antioxidants are added to foods containing fat, especially meats, with the purpose of delaying oxidative processes that result in undesirable sensorial changes, decreased shelf life and nutritional value and the formation of secondary compounds potentially harmful to health [7,8]. However, data in the literature have associated the synthetic antioxidants BHT (butylated hydroxytoluene) and BHA (butylated hydroxyanisole) with possible deleterious health effects, as they have been reported as potential tumor promoters, following the chronic administration of these compounds to animals [9,10]. This has motivated the replacement of synthetic antioxidants with natural antioxidants extracted from foodstuffs [11,12].

Betanin can be used as a powerful antioxidant in the food industry in extract or powder form, and is also applied as a natural pigment. Its antioxidant activity in biological lipid environments has been demonstrated in human macromolecules, such as low density lipoproteins, membranes and whole cells [13]. Furthermore, betanin has attracted attention due to its anti-inflammatory and hepatic protective functions in human cells. This compound is able to modulate redox-mediated signal transduction pathways involved in inflammation responses in cultured endothelia cells, and has also displayed antiproliferative effects on human tumor cell lines [14,15]. In both healthy and tumoral human hepatic cell lines, betanin can induce the translocation of the erythroid 2-related factor 2 (Nrf2) antioxidant response element (ARE) from the cytosol to the nuclear compartment, which controls the mRNA and protein levels of detoxifying/antioxidant enzymes, including GSTP, GSTM, GSTT, GSTA (glutathione *S*-transferases), NQO1 (NAD(P)H quinone dehydrogenase 1) and HO– (heme oxygenase-1), in these cells, exerting hepatoprotective and anticarcinogenic effects [16].

Several betanin purification techniques have been reported, involving distinct steps and methodologies in order to purify this compound from vegetal sources, including complex food matrices such as beetroot. Among the methods employed for betanin purification, high-performance liquid chromatography (HPLC) and other chromatographic methods using reverse phase columns seem to provide the best balance between speed and efficiency [17]. However, no other studies have evaluated the stability of this molecule during storage conditions and its antioxidant capability after purification and during storage with the aim of use as a food additive.

In this context, the aim of the present study was to optimize a methodology applied for betanin purification in large amounts, using fresh juice obtained from red beetroot (*Beta vulgaris* L. species). In addition, the chemical stability and bioactivity of the purified molecule were also assessed, through two different viewpoints: (i) as a food preservative and (ii) as a *in natura* or processed food matrix component after consumption and simulated gastrointestinal and ex vivo colon fermentation processes.

## 2. Results and Discussion

### 2.1. Betanin Purification

The chromatographic profile of betanin and isobetanin in fresh beetroot juice was compared to a commercial standard (Figure 1A,B). In the fresh juice chromatogram, betanin corresponds to the major peak (RT = 18.17 min), followed by its isomer, isobetanin (RT = 19.27 min). Betanin is found in large amounts relative to its isobetanin isomer in fresh beetroot juice, which is similar to the ratio between betanin and isobetanin previously reported by Gonçalves et al. [17]. The betanin concentrations in fresh juice and in purified samples were calculated in comparison to a standard curve, described by the equation y = 5077078x − 370531 (R^2^ = 0.9993). Linearity was obtained at betanin concentration ranging from 0 to 500 mg∙mL^−1^. After purification an HPLC-diode array detector (DAD) analysis indicated a single peak at the maximum absorption λ (535–540 nm), characteristic for betacyanins (Figure 1C). Purified betanin can be preserved at 4 °C for at least 20 days and for 275 days if maintained at −30 °C (Figure 1D). On the contrary of what has been described previously [17], betanin preparations turned brown during shelf life, due to the possible action of polyphenol oxidases (PPO enzymes) in the present study, whereas no sign of degradation was observed in the purified betanin. Temperature control, pH and non-exposure to light throughout the purification process reduced the chances of sample decomposition. Such precautions may have had a positive influence on the stability of the molecule, as demonstrated later in the stability assessment.

Purification technique yield is one of the most economically important aspects when obtaining natural food additives. The betanin extraction and purification yield from fresh beetroot juice and purified betanin recovered after chromatographic separation and mobile phase evaporation was calculated from an initial 500 g beetroot mass. The betanin concentration in juice was of 1.19 g mL^−1^ and the amount of purified betanin recovered after purification by chromatographic separation and mobile phase evaporation was of 48 mg∙mL^−1^ corresponding to 4% of the initial betanin concentration.

As noted, in Figure 1C the beginning of betanin isomerization to isobetanin occurred, culminating with co-elution and a small base widening. The analytical column used for the separation and the elution conditions were not able to clearly separate the peaks in the initial isomerization stage, but the presence of the already formed isobetanin was well-evidenced and the peaks were clearly separated during storage (Figure 1D). It is noteworthy that betanin isomerization can be considered a structural modification rather than a degradation reaction. Isobetanin (2*S*/15*R*) differs from betanin (2*S*/15*S*) by the spatial conformation of the carbonyl group at carbon 15, but exhibits similar chromatic properties to betanin with no observable color change [18].

### 2.2. HPLC-ESI(+)-MS/MS Analysis

HPLC-purified betanin identification was performed by mass spectrometry, as displayed in Figure 2A,B. The pseudomolecular ion *m/z* 551.16 corresponding to betanin (Figure 2A) and its characteristic fragmentation (Figure 2B), the ion *m*/*z* 389.11 corresponding to its aglycone form, betanidin, a precursor structure of betacyanins (Appendix A), were observed, corroborating previous findings reported by Gonçalves et al. [17] and Netzel et al. [20].

It can be suggested that the *m*/*z* 637 adduct may be the result of the condensation of the formic acid (molecular weight 46) used in betanin purification and a decarboxylated betanin derivative, forming 6′-*O*-malonyl-2-decarboxyl-betanin (molecular weight 592), a decarboxylated betanin derivative [21]. The mass spectrometry analysis confirmed that the purification procedure was successful in isolating betanin in its purified form.

### 2.3. Storage Stability

Betanin was stable for over 275 days (9 months) when stored at −30 °C at pH 7 (Figure 1D) and at least for 20 days when stored at 4 °C. No significant loss of the betanin samples was observed before and after frozen storage (21.30 ± 1.98 mg∙mL^−1^ versus 17.23 ± 4.82 mg∙mL^−1^, *p* < 0.001, respectively). The initial concentration of purified betanin samples during storage at refrigerator temperature was unaltered during the first 20 days (12.13 ± 0.70 mg∙mL^−1^ versus 13.51 ± 1.01 mg∙mL^−1^, *p* < 0.001) but on the 27th day, a loss of 25% of the initial betanin concentration was observed (9.72 ± 0.60 mg∙mL^−1^).

Most betalains, including betanin, are under-utilized as colorants in processed foods due to reports concerning poor stability compared with the shelf-life of foods. In standard storage conditions betacyanin stability in spray dried beetroot powder was reported by Kaimainen et al. [22], who assessed the product stability by HPLC at 535 nm and pH 5 during storage for 25 weeks at different temperatures, including frozen (−20 °C) and chilled (4 °C). Betacyanins in beetroot powder remained unchanged during 4 months under freezing. However, stability under refrigeration was not well established. It is noteworthy that in that study, beside the addition of sweeteners, betanin was not in its purified form, but rather protected by the food matrix that naturally contains antioxidants and chelating agents, which may exert protective effects on the chemical structure of betanin maintained under freezing conditions [18]. Factors such as temperature, pH, type of buffer solution and the presence or absence of oxygen can affect betanin stability during storage. Betanine degradation results in the formation of betalamic acid and cyclo-dopa-5-*O*-glycoside. However, betanin can also display the ability to degrade and regenerate continuously during storage, as the reaction is reversible, thus maintaining betanin concentrations [23]. This continuous betanin regeneration capacity during the storage process is still not well-elucidated in the literature [2,23,24]. Therefore, the stability results found in the present study indicate that betanin, if used as an additive in refrigerated or frozen foods, would remain stable during the time and temperature recommended by the legislation as ideal for preserving meats and meat-derivatives from 6 to 12 months at temperatures below −18 °C and for 5 days at 4 °C [25].

### 2.4. Lipid Peroxidation Inhibition in Meat Matrices

Betanin’s ability to inhibit lipid peroxidation process in meat was assessed by thiobarbituric acid reactive substance (TBARS) determination. Control samples (without antioxidants) showed the highest malondialdehyde (MDA) concentrations when compared to samples treated with betanin, BHA or BHT during 9 days of storage, determined on the 3rd, 6th and 9th days. Betanin 2% was equally effective in inhibiting lipid peroxidation when compared to the synthetic antioxidants BHA and BHT on the 3th and 6th day of storage (*p* < 0.05). Although the amounts of MDA in meat samples treated with betanin 2% (*w*/*w*) (5.07 ± 0.03 mg∙kg^−1^) were 22% and 16% higher than those found in samples treated with BHA (4.47 ± 0.10 mg∙kg^−1^) and BHT (4.28 ± 0.38 mg∙kg^−1^) on the 9th day of storage, the meat samples treated with betanin 2% (*w*/*w*) still presented lower MDA concentrations than the samples with no preservative addition (7.15 ± 0.07 mg∙kg^−1^ vs 5.07 ± 0.03 mg∙kg^−1^) (Figure 3).

Few studies assessing the effect of the addition of beetroot or beetroot-extracted compounds on the oxidative stability of foods susceptible to lipid oxidation are available. The effect of beetroot inclusion as a mayonnaise ingredient promoted a higher inhibitory effect on lipid oxidation compared to the commercial antioxidant [26]. Contradictory results, however, were observed in fermented meat sausages, since no beetroot effect was observed [27].

To the best of our knowledge, the present study is the first to evaluate the use of purified betanin as a natural antioxidant in food matrices. Lipid oxidation is one of the main factors affecting food quality and is directly related to nutritional value and sensorial characteristics. The present study indicates that betanin used as an additive at the concentration of 2% (*w*/*w*) is a potential substitute for synthetic antioxidants in the preservation of refrigerated meat. Furthermore, betanin can exert its maximum protective effect against lipid oxidation for 6 days exceeding the 5-day shelf-life recommended for refrigerated meats [25].

### 2.5. Betanin Chemical Stability during In Vitro Simulated Gastrointestinal Digestion

A 23 mg∙mL^−1^ dose was used to assess betanin bioavailability during in vitro human simulated gastrointestinal digestion by continuous multistage steps. A small loss was observed in betanin content after the oral phase digestion. However, more important decreases were observed after the gastric simulated digestion, reaching 65% of the initial sample content, and lowering to 46%, after small intestine simulated digestion (Table 1). No betanin was detected after the ex vivo colon fermentation assay, where the remaining betanin recovered at the end of the in vitro simulated gastrointestinal digestion, corresponding to 54% of the original sample, was assayed by ex vivo colon fermentation (Appendix A).

Only a few studies presenting limitations are available on the chemical stability of the purified betanin in in vitro simulated digestion through the gastrointestinal tract. In a previous report, betanin was degraded by 75% and 35% after the gastric and intestinal phases, respectively [28]. However, the sample was added directly to each fluid—gastric or intestinal, generating no information about physiological digestion in the digestive tract. In the present study, an in vitro simulated digestion was conducted sequentially, where betanin was added to simulated oral fluid and digestion was observed by a sequential in vitro system, transferring aliquots to the simulated gastric fluid, resulting in a decrease of 35% after gastric digestion. This 35% decrease in betanin contents observed after gastric digestion is due to its impaired stability at acidic pH 2. It is known that betalains exhibit stability at pH ranging from 3 and 7 [18]. A significant decrease in betacyanin stability in a solution containing hydrochloric acid at pH 2.0 at 37 °C was observed, whereas betacyanins maintained at pH 4.7 were less susceptible to degradation [29]. In acid pH, the betanin structure can be degraded in C-17 decarboxylation, dehydrogenation and cleavage of betalamic acid and cyclo-Dopa-5-*O*-β-glycoside [2,30]. Herein, in addition to exposure to acidic pH, the absence of the food matrix may have exacerbated the betanin susceptibility to gastric fluid degradation, since it has been previously demonstrated that betanin and its isobetanin isomer (unpurified) can be protected from stomach digestion by the food matrix [31].

In addition, an overall decrease in betanin content to approximately 46% was noted when the simulated gastric fluid digestion was performed, followed by an 11% decrease during intestinal digestion. Besides the absence of the protective effect of the food matrix, the influence of pH on betanin stability is reinforced by the data reported herein, comparing the percentage of loss in the gastric phase to the intestinal phase (35% vs 11%). Small intestine pH is around 6.5, matching the reported pH-range of betanin stability and corroborating the lower betanin degradation at the small intestine [31].

Several polyphenols are described as reaching the large intestine, where they are absorbed following metabolization by colon bacteria consortia [32]. In the present study, betanin was detectable by HPLC at the 536–540 nm range, while no other metabolite derived from its biosynthetic conversion was observed. Although betanin was shown to be stable at 30 °C and 4 °C, the 24 h exposure to 37 °C and environmental conditions in the colon lumen may still promote its degradation. In a murine model study, only 3% of betanin administered to animals were recovered in feces after 24 h, indicating that colon absorption is not likely [28]. Permeability and solubility are important barriers concerning colon absorption in humans that, alongside the metabolic activity of bacterial consortia and the mild temperatures within the large intestine lumen point to improbable colon absorption and/or maintenance of the chemical stability of the betanin molecule.

### 2.6. Betanin Antioxidant Activity throughout Simulated Human Gastrointestinal Digestion

Betanin in its purified form was able to inhibit the OH-radical in the total antioxidant potential (TAP) assay. The OH-radical is considered the most reactive oxidant in living organisms, generated by the Fenton reaction [33] (Appendix A). In the ferric reducing ability of plasma (FRAP) assay, betanin was effective in reducing the ferric ion of the tripyridyltriazine complex (Fe^3+^-TPTZ) to the ferrous ion (Fe^2+^-TPTZ), reflecting its ability to donate electrons and reduce reactive species. In addition, betanin was effective in reducing the ABTS^+^ radical, as observed in the trolox equivalent antioxidant capacity (TEAC) and oxygen radical antioxidant capacity (ORAC) assays (Table 2). In addition, betanin showed a high TAP value after both oral and intestinal digestion.

The high antioxidant activity of betanin is well-documented [34]. However, to promote health beneficial in human beings, the chemical structure of the ingested betanin and its antioxidant properties should be maintained in the gastrointestinal absorption site. The antioxidant activities of purified betanin following the final sequential digestive process through the gastrointestinal apparatus increased when compared to pre-digested samples, as demonstrated in the FRAP and TAP assays (Table 2).

When assessing each digestion fluid, the antioxidant activity of betanin evaluated by the FRAP assay was increased in the post-oral digestion assays. The FRAP assay increments can be attributed to natural antioxidants originally present in human saliva [35]. After small intestine digestion, purified betanin increased or maintained the antioxidant activities evaluated by the TAP, FRAP, TEAC and ORAC assays when compared to the oral digestion processes (Table 2).

Additionally, the pH of the different fluids in the human body undergoes variations, influencing the stability and bioactivity of betanin in the different digestive tract compartments. Salivary fluid present a pH of about 7.4, whereas stomach is maintained between 1.5 and 3.5, while the abdominal cavities, including the small and large intestine, display a pH of 7.4.

The antioxidant ability of betanin was reduced in the acidic pH of the gastric fluid, evidencing the pH-dependence of the free radical-scavenging activity of betanin, but, countering its antioxidant activity in the alkaline environment of the small intestine lumen, where the antioxidant activity assessed by all assays was either increased or maintained at the same level of the oral fluid, as mentioned previously (Table 2). The alternate antioxidant activity of betanin between low and high levels in the simulated gastric compartment and in the small intestine indicates that the chemical structure of betanin was maintained unaltered following acidic pH exposure during the simulated gastric digestion, but should be attributed to the protonation of the betanin molecule, favoring the maintenance of free radical-scavenging activity until absorption in the gastrointestinal tract.

Changes in pH fluids induce changes in the chemical structure of betanin along the gastroenteric apparatus. Betanin presents its cationic form at pH < 2, zwitterionic form at pH = 2, anionic mono at 2 < pH < 3 with deprotonated C2-COOH and C15-COOH groups, anionic bis at 3.5 < pH < 7 with C2-COOH, C15-COOH and C17-COOH groups deprotonated and anionic tris at 7.5 < pH < 9 with all carboxyl deprotonated groups, in addition to the C6-OH group on the phenolic ring (Appendix A) [36]. The increase in betanin bioactivity, when in an alkaline pH environment, as found in the simulated small intestine fluid, can be ascribed to its ability to donate H^+^ and electrons when altering from the cationic to the mono, bis and tri deprotonated states. The free radical scavenging activity of betanin at different pH (from 2 to 9) was previously determined through the TEAC assay, phenolic O–H homolytic bond dissociation energy (OH BDE), ionization potential (IP) and deprotonation energy (DE) [37]. The TEAC assay indicated that the antioxidant activity of betanin is dependent on pH, and very high above pH > 4. Moreover, with the gradual increase of the deprotonation of the betanin molecule (mono, bi- and tri-deprotonated) according to increasing pH, BDE and PI values decreased. This implies that, at slightly alkaline pH, betanin becomes a better hydrogen and electron donor, increasing its radical-scavenger ability as observed in the antioxidant assays in the simulated small intestine fluid when compared to the gastric fluid.

Although almost 46% (11 mg) of betanin content were chemically modified in the gastric tract, the antioxidant power of the remaining betanin, 54% of the original amount, corresponding to 12 mg found in the small intestine fluid (21 µmol) seems to be enough to promote lipid oxidation inhibition, since betanin in the range of 0.3–1.9 µmol has been found to inhibit lipid peroxidation in biological membranes, in a linoleate emulsion catalyzed by the “free iron” redox cycle, in H_2_O_2_-activated metmyoglobin and in lipoxygenase activity [38].

## 3. Material and Methods

### 3.1. Standards and Reagents

The betanin standard (C_24_H_26_N2O_13_), sulfuric acid (H_2_SO_4_), boric acid (H_3_BO_3_), formic acid (CH_2_O_2_), hydrochloric acid (HCL), terephthalic acid (TPA, C_8_H_6_O_4_), ethylenediaminetetraacetic acid (EDTA), 6-hydroxy-2-5-7-8-tetramethylchromo-2-carboxylic acid (Trolox), ascorbic acid (C_6_H_8_O_6_), sodium hydroxide (NaOH), potassium permanganate (KMnO_4_), hydrogen peroxide (H_2_O_2_), potassium sulphate (K_2_SO_4_), ferrous sulphate (FeSO_4_), methyl red, bromocresol green, petroleum ether, anhydrous sodium acetate (CH_3_COONa), tetrabutylammonium perchlorate (C_16_H_36_N.H_2_PO_4_), vanillin, tripyridyltriazine (TPTZ, C_18_H_12_N_6_), iron chloride (FeCl_3_), dibasic sodium phosphate (NaH_2_PO_4_.H_2_O), monobasic sodium phosphate (NaH_2_PO_4_.H_2_O), sodium chloride (NaCl), anhydrous monobasic sodium phosphate (Na_2_HPO_4_), sodium bicarbonate (NaHCO_3_) C-211,2,2′-Azobis (2-methylpropionamidine), dihydrochloride (AAPH), 2,2′-azinobis [3-ethylbenzothiazoline-6-sulfonic acid]-diammonium salt (ABTS, C_18_H_24_N_6_O_6_S_4_), sodium fluorescein (C_20_H_12_O_5_), potassium hydroxide (KOH), ammonium thiocyanate (NH_4_SCN), trichloroacetic acid (Cl_3_CCOOH) were purchased from Sigma-Aldrich Chemical Co. (São Paulo, SP, Brazil). Methanol (MeOH), ethanol, acetone, and acetonitrile were purchased from Tedia Company Inc. (Rio de Janeiro, RJ, Brazil). Buthylated hydroxyanisole (BHA), butylated hydroxytoluene (BHT), 1,1,3,3-tetramethoxypropane ((CH_3_O)_2_CHCH_2_CH(OCH_3_)_2_) and 2-thiobarbituric acid (C_4_H_4_N_2_O_2_S) were purchased from Sigma-Aldrich Co. HPLC grade Milli-Q water (Merck Millipore, Burlington, MA, USA) was used throughout the experiments.

### 3.2. Betanin Purification

#### 3.2.1. Sample Preparation

Red beetroot was peeled, sliced and homogenized using a centrifuge food processor EC 700 (Black and Decker, São Paulo, Brazil). The homogenates were centrifuged at 15,000× *g* for 30 min at 25 °C and filtered through a PTFE filter membrane 25 mm, pore size 0.45 µm (Merck-Millipore). The supernatants (4 mL) were concentrated under reduced pressure (18 mbar, 25 °C) and resuspended in 2 mL deionized water.

#### 3.2.2. HPLC Betanin Purification

Concentrated beetroot juice was purified by RP-HPLC. The HPLC apparatus consisted in an LC-20A Prominence, (Shimadzu^®^, Kyoto, Japan) equipped with a quaternary pump and a DAD model SPD-M20A (Shimadzu^®^, Kyoto, Japan). A 15 µm Phenomenex C18 column (250 × 21.2 mm I.D., Torrance, California, USA) connected to an FRC-10A fraction collector (Shimadzu^®^) was used in the semi-preparative HPLC. The elution conditions were performed according to Cai et al. [39] with modifications. Solvent A was 1% formic acid, and solvent B was 80% methanol at a linear gradient (0–25 min, 11–55%). The injection volume was 100 μL and a flow rate of 5.5 mLmin^−1^ was used. Separations were monitored at 536 nm and, after purification, magenta fractions, containing betanin, were concentrated by a rotary evaporator (Rotavapor^®^ R-215, Buchi, São Paulo, Brazil) at 24 °C, 150 rpm and a water bath at 40 °C. The extracts were then suspended in 1 mL deionized water and stored at −30 °C under an N_2_ atmosphere for further analysis. The purified betanin was analyzed using a Nucleosil 100-C18 column (250 × 4.6 mm I.D., 5 μm) with 30 µL injection volume and a flow rate of 1.0 mL min^−1^. The mobile phase and gradient conditions were similar to the purification step and betanin concentrations were quantified in comparison to a betanin standard solution (Sigma-Aldrich Co.).

### 3.3. Betanin Identification by Liquid Chromatography Positive Ion Electrospray Ionization Tandem Mass Spectrometry (LC-ESI(+)-MS/MS)

Mass spectrometry was performed as described by Gonçalves et al. [17]. The RP-HPLC purified fraction was ionized in the positive mode and ions were monitored in the full scan mode (range of *m*/*z* 50–1500). The ESI(+)-MS/MS analysis was carried out on a Bruker Esquire 3000 Plus Ion Trap Mass Spectrometer (Bruker Co., Billerica, MA, USA) equipped with an electrospray source in the positive ion mode. Nitrogen was used as the nebulizing (45 psi) and drying gas (6 L∙min^−1^, 300 °C) and helium as the buffer gas (4 × 10^−6^ mbar). The high capillary voltage was set to 3500 V. To avoid space–charge effects, smart ion charge control (ICC) was set to an arbitrary value of 50.000. Betanin identification was based on its mass (550 g∙mol^−1^) and by similarity with the commercial standard and literature-available spectra [39].

### 3.4. Storage Stability

The stability of purified betanin during refrigeration (4 °C) and freezing (−30 °C) was evaluated by RP-HPLC-DAD (Shimadzu^®^, Kyoto, Japan), monitoring changes in the area under the chromatogram peak obtained at 536 nm, in similar conditions as those described for the betanin analysis.

### 3.5. Betanin Ability to Inhibit Lipid Peroxidation in Meat

Betanin ability to inhibit lipid peroxidation was evaluated by MDA determination in meat TBARS assay, as described previously [40] with modifications. A sample of ground pork loin (500 g) from a local butcher shop in Rio de Janeiro, Brazil and divided into 4 portions and treated as follow: (i) ground pork loin non-treated by antioxidants; (ii) ground pork loin treated with betanin (2%; *w*/*w*); (iii) ground pork loin treated with BHT (0.01%); (iv) ground pork loin treated with BHA (0.01%). MDA extraction was performed in 3.0 g of each meat sample homogenized with 9 mL of 7.5% TCA. The homogenate was centrifuged at 3000× *g* for 15 min at 25 °C and filtered through Whatman n° 4 paper (Merck Millipore Co). TMP (the MDA standard) at 3.2 mM in 0.1 M HCl (stock solution) was kept for 2 h at room temperature in the dark. After hydrolysis, the TMP solution was diluted with 7.5% TCA to the concentrations of 1, 2, 4, 8, 16 and 32 µM. After, 1 mL of MDA at different concentrations or 7.5% TCA solution (blank) was transferred into a screw-cap tube and 1 mL of 20 mM TBA solution was added. The tubes were heated in a boiling water bath at 90 °C for 30 min and cooled in tap water for 10 min. Absorbance of the MDA-TBA adducts were measured at 532 nm on a spectrophotometer DU^®^530 (Beckman Coulter Inc., Brea, CA, USA). Because betanin absorbs light in the range of 530–540 nm, additional blanks containing betanin (1 or 2%), TCA or TBA (no meat) were used to correct the overestimation of the TBA-MDA adduct absorbance. The concentration of MDA was expressed in mg of MDA per kg of meat (mg of MDA∙kg^−1^ meat) at each treatment along the 9 days storage at 4 °C.

### 3.6. TAP Determination

Betanin samples were analyzed as described previously [41]. Samples were diluted (1:10) and centrifuged at 4500× g for 10 min, and the supernatants were then filtered through 0.45 µm cellulose membranes (Merck Millipore Co). The resulting samples were transferred to amber vials and incubated at 37 °C for 10 min with a solution containing 1 mM Fe^2+^, 10 mM H_2_O_2_ and 1 mM terephthalic acid (TPA) in 50 mM phosphate buffer pH 7.4. Hydroxyterephthalic acids (HTPA) were detected by HPLC. TAP measurements were obtained by the difference between the chromatogram surface area generated in the Fenton reaction with and without the sample.

### 3.7. Antioxidant Activity Determination by Different Assays

#### 3.7.1. FRAP Determination

FRAP assays were performed using a modification of the method described by Benzie and Strain [42]. Betanin samples were diluted (1:10) and then mixed thoroughly with the FRAP reagent. Standard FeSO_4_ solutions were used and absorbances at 593 nm were determined on a V–530 UV/VIS spectrophotometer (Jasco^®^, Easton, PA, USA). The FRAP results for each sample were evaluated in triplicate and expressed as µmol of Fe^2+^∙L^−1^.

#### 3.7.2. TEAC Determination

TEAC assays were performed using a modification of the method described by Re et al. [43]. The ABTS radical cation (ABTS^•+^) was generated by chemical reaction of ABTS with K_2_S_2_O_8_ in the dark at room temperature for 12–16 h. Each betanin sample was mixed with the ABTS^•+^ reagent and absorbances at 720 nm were determined using a V–530 UV/VIS spectrophotometer (Jasco^®^). TEAC results were determined in triplicate and were associated to the ABTS^•+^ inhibition percentage by antioxidants present in the samples. The TEAC results for each beetroot sample were evaluated in triplicate and expressed as µmol of Trolox∙L^−1^.

#### 3.7.3. ORAC Determination

The ORAC assay was performed according to Zuleta et al. [44], with modifications. Sample absorbances were determined on a Wallac 1420 VICTOR multilabel counter (Perkin–Elmer Inc, Waltham, MA, USA) with fluorescence filters at an excitation wavelength of 485 nm and emission wavelength of 535 nm. A fluorescein stock solution was prepared by weighing 3 mg of fluorescein followed by dissolution in 100 mL of phosphate buffer (75 mM, pH 7.4). The fluorescein stock solution was stored in complete darkness under refrigeration. The fluorescein working solution (78 nM) was prepared daily by dilution of 0.100 mL of the fluorescein stock solution in 100 mL of phosphate buffer. The AAPH radical (221 mM) was prepared daily by mixing 600 mg of AAPH in 10 mL phosphate buffer. A 25 µM Trolox solution was used as reference standard, prepared daily in phosphate buffer from a 4 mM stock standard solution kept in a freezer at 20 °C. A total of 100 μL of fluorescein (78 nM) and 100 μL of the samples, blanks (phosphate buffer), or standards (25 µM of Trolox) were added to each well, followed by 50 μL of AAPH (221 mM). ORAC values, expressed as μM Trolox equivalents were calculated by applying the following formula:ORAC (µM Trolox equivalents) = C_Trolox_·(AUC_Sample_ − AUC_Blank_)·*k*
(AUC_Trolox_ − AUC_Blank_)
where C_Trolox_ is the Trolox concentration (µM), *k* is the sample dilution factor, and AUC is the area below the fluorescence decay curve of the samples, blanks and Trolox, respectively, calculated using the GraphPad Prism v.5 software package (GraphPad Software Inc., San Diego, CA, USA). ORAC determinations were performed in triplicate and expressed as mmol Trolox equivalents·100 g^−1^.

### 3.8. Simulated Betanin In Vitro Human Gastrointestinal Digestion and Ex Vivo Colon Fermentation (Appendix A)

Betanin concentrations after in vitro oral, gastric and small intestine digestion and ex vivo colon fermentation were evaluated by RP-HPLC while antioxidant activity was evaluated by TAP, FRAP, TEAC and ORAC assays, as described previously. Samples were analyzed in triplicate.

The in vitro human simulated gastrointestinal digestion, including the oral, gastric and small intestine phases, was performed according to Oomen et al., [45] and Sagratini et al. [46], with modifications. For the OD simulation, 1 mL betanin aliquots at 23 mg·mL^−1^ were placed in a glass jar followed by the addition of 3 mL of human saliva, and incubated at 37 °C for 1 min under orbital agitation at 260 rpm in a Sorvall ST 16R centrifuge (Thermo Scientific^TM^, Waltham, MA, USA) to complete the OD.

A 2.5 mL aliquot of artificial gastric fluid containing 2.75 g of NaCl, 0.27 g of NaH_2_PO_4_, 0.82 g of KCl, 0.42 g of CaCl_2_, 0.31 g of NH_4_Cl, 0.65 g of glucose, 0.085 g of urea, 3.0 g of mucine, 2.64 g of swine gastric pepsin, 1.0 g of bovine albumin, 8.3 mL of HCl was added to the oral fluid sample to a final volume of 500 mL and the pH was adjusted to 2.0 with 5 M HCl. The glass jars were then resealed with a rubber septum and the atmosphere was saturated with N_2_ and incubated at 37 °C for 2 h under orbital shaking at 260 rpm to complete the GD.

The gastric fluid had its pH adjusted to 6.0 with NaHCO_3_ and 2.0 mL of artificial small intestine fluid containing 6.75 g of NaCl, 0.517 g of KCl, 0.205 g of CaCl_2_, 3.99 g of NaHCO_3_, 0.06 g of KH_2_PO_4_, 0.0375 g of MgCl_2_, 0.1375 g of urea, 25.0 g of swine bile, 4.0 g of swine pancreatin, 1.2 g of albumin bovine and 0.185 mL HCl were added to a final volume of 500 mL. The glass jars were then resealed and the atmosphere was saturated with N_2_ and incubated at 37 °C for 2 h under orbital shaking at 260 rpm to complete the ID. At the end of each simulated gastrointestinal step (oral, gastric and small intestine), aliquots were collected and centrifuged (3000× *g*, 15 min, 25 °C). The supernatants were then filtered through 0.45 and 0.22 μm membranes, followed by Amicon ultra filtration using 10 kDa cut-off membranes.

#### Ex vivo Colon Fermentation

The ex vivo colon fermentation assay was performed according to Hu et al. [47] with modifications, in accordance to the ethical standards of the declaration of Helsinki after approval by the Hospital Universitário Clementino Fraga Filho/Universidade Federal do Rio de Janeiro Education and Research Committee, under No. 512.84.

The ex vivo assay was performed using fresh feces donated by seven healthy volunteers (4 men and 3 women), recruited according to the following criteria: age between 18 and 50, eutrophic (BMI between 18.5 and 24.9 kg m^2^), absence of gastrointestinal diseases, displaying one bowel movement every two days and up two bowel movements per day, with no medication and/or food supplements used 90 days prior to the feces collection.

The feces were homogenized in a nutrient-rich medium (0.5 g·10 mL^−1^), prepared according to McDonald et al. [48], where the medium was autoclaved and saturated with CO_2_ in an anaerobic chamber for 48 h. A 5 mL aliquot of this mixture was added to the digested material after the in vitro gastrointestinal digestion. The mixture was then incubated at 37 °C, under orbital shaking at 50 rpm for 48 h. The ex vivo colon fermentation assay was independently repeated three times.

### 3.9. Statistical Analyses

A one-way analysis of variance (ANOVA) with repeated measurements were performed to identify differences in TAP and antioxidant activities (FRAP, TEAC and ORAC) before and after the simulated gastrointestinal digestion (pre-digestion, oral, gastric, small intestine and colon phases). In addition, a two-way analysis of variance (ANOVA) with repeated measurements was performed to identify differences in MDA concentrations in the lipid peroxidation assay between each type of antioxidant and between each experiment day, evaluated by TBARS assay. When a significant *F* was found, an additional *post hoc* analysis was performed by a Bonferroni correction. Data were expressed as means ± standard deviation (SD). The statistical analyses were performed using Graphpad Prism software version 5 for Windows^®^ (GraphPad Software, San Diego, CA, USA).

## 4. Conclusions

The decomposition of food components and bioactive or additive compounds along the gastrointestinal trait is a helpful tool to assess their potential positive and negative effects on health, as well as to evaluate possible toxicity and/or safety usage. The prediction of the in vivo absorption of such compounds can aid in compound usage regulation, establishing safety dosages in food.

Betanin in its purified form can be very stable during storage at low temperature and alkaline pH, so it may be useful as food colorant and antioxidant additive in meat and meat derivatives as a substitute for synthetic antioxidants. According to the chemical stability at both refrigerator and freezer temperatures, betanin can be used as a food colorant or preservative in almost all foodstuffs, including those stored at −30 °C, such as bacon, sausages, vegetables, ham, corned beef, ice cream and sherbet (www.fda.gov/food/guidance/regulation) and those preserved at 4–8 °C, including minced meat and fresh meat, yogurts and desserts, since their shelf life is lower than the 20 days in which betanin is stable.

In addition, betanin maintained its bioactivity during the simulated digestive process, presenting high TAP, FRAP, TEAC and ORAC values in the intestinal phase. Although the exact parts of the gastrointestinal tract in where betanin is absorbed still require elucidation, it can be suggested that absorption may occur in the small intestine. The chemical integrity of betanin and its antioxidant activity can be considered potential aid against diseases caused by oxidative stress.

These novel findings reinforce the importance of the regular uptake of red beetroot and its derivative products. The formulation of new dietary supplements or processed foods can include purified betanin, not only as a natural food colorant or preservative, but also as a bioactive compound that may act as an adjuvant in the treatment and prevention of chronic diseases related to oxidative stress in humans.

## Figures and Tables

**Figure 1 molecules-24-00458-f001:**
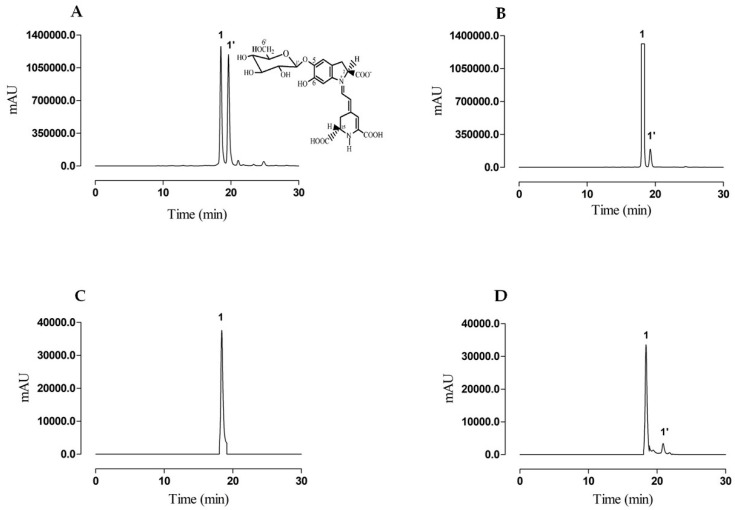
Betanin separation by high-performance liquid chromatography diode array detector (HPLC-DAD) monitored at 536 nm. (**A**) Betanin standard chromatographed in the analytical HPLC column, (**B**) fresh beetroot juice sample chromatographed in semi-preparative HPLC, (**C**) betanin purified by semi-preparative HPLC and separated using an analytical HPLC column and (**D**) betanin evaluated after 275 days of freezing and chromatographed using an analytical HPLC column. Betanin (peak 1) and isobetanin (peak 1’). The betanin chemical structure from red beet was reproduced from Cai et al. [19].

**Figure 2 molecules-24-00458-f002:**
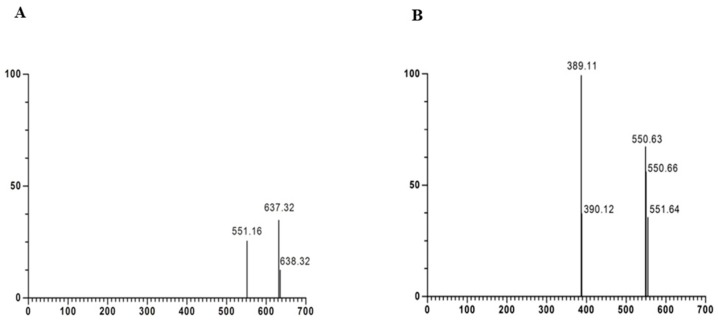
Identification of purified betanin by HPLC-ESI(+)-MS/MS. (**A**) betanin *m*/*z* 551 [M + H]^+^, (**B**) fragmentation of purified betanin *m*/*z* from the MS/MS of 551 [M + H]^+^.

**Figure 3 molecules-24-00458-f003:**
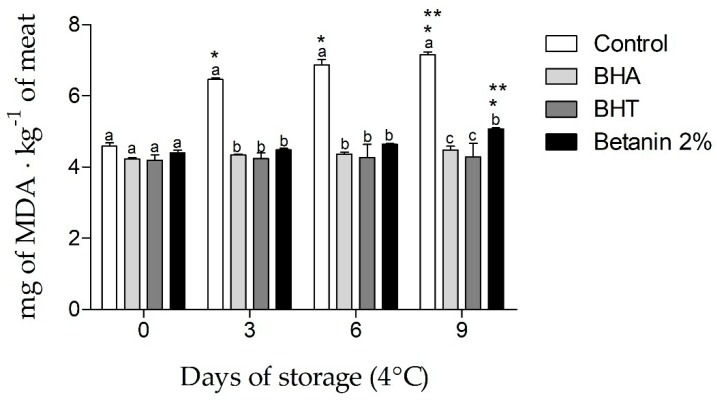
Lipid oxidation in ground pork loin evaluated by the production of malondialdehyde (MDA) during 9 days of storage at 4 °C. Control H_2_O-DD, BHA (buthylated hydroxyanisole), BHT (butylated hydroxytoluene), Betanin 2% (*w*/*w*). Data are expressed as the means ± SD of three independent determinations. Different letters indicate differences between days at a significance level of *p* < 0.01. The symbol * (*p* < 0.05) indicates differences compared to day 0. The symbol ** (*p* < 0.05) indicates differences compared to day 3.

**Table 1 molecules-24-00458-t001:** Betanin concentrations during in vitro simulated gastrointestinal digestion.

	Pre-Digestion	Oral Fluid	Gastric Fluid	Small Intestine Fluid	Colon Fermentation Fluid
Betanin content (mg∙mL^−1^)	23.05 ± 0.61 ^a^	21.44 ± 2.03 ^a^	14.84 ± 0.11 ^b^	12.42 ± 0.01 ^c^	0.0
Loss (mg∙mL^−1^) and loss percentage after pre-digestion	-	1.6 (≈7%)	8.2 (≈35%)	10.6 (≈46%)	-

Betanin availability was determined by reverse phase high-performance liquid chromatography diode array detector (RP-HPLC-DAD), assessed through changes in the peak area determined at 536 nm. In vitro human gastrointestinal digestion was sequentially simulated and samples were harvested at each phase. The ex vivo colon assay was performed incubating the digested material obtained after the entire in vitro gastrointestinal digestion with fresh feces donated by seven healthy volunteers. Data are expressed as the means ± SD of three independent experiments. Different letters in the same line indicate significant differences between samples (*p* < 0.01).

**Table 2 molecules-24-00458-t002:** Total betanin antioxidant potential and antioxidant activity pre and post in vitro simulated human gastrointestinal digestion.

		TAP (%)	FRAP µmoL (Fe^2+^∙L^−1^)	TEAC µmoL (Trolox∙L^−1^)	ORAC µmoL (Trolox∙L^−1^)
Pre-digestion	Betanin	75.42 ± 5.91 ^b^	518.31 ± 3.31 ^c^	3932.02 ± 94.42 ^a^	1992.44 ± 214.31 ^ab^
Post-digestion	Oral fluid	80.71 ± 0.92 ^b^	585.82 ± 13.23 ^b^	4964.03 ± 5.31 ^a^	2217.53 ± 10.31 ^a^
	Gastric fluid	55.11 ± 9.23 ^c^	400.02 ± 12.43 ^d^	1382.94 ± 4.91 ^b^	1475.41 ± 18.73 ^c^
	Small intestine fluid	96.63 ± 0.61 ^a^	1053.81 ± 164.64 ^a^	4312.71 ± 651.81 ^a^	2199.71 ± 19.75 ^a^

Betanin antioxidant potential and antioxidant activity were evaluated before and after the simulated human gastrointestinal digestion using different assays, namely FRAP, TEAC and ORAC. Data are expressed as the means ± SD from three independent experiments. Different letters in the same column indicate difference at a significance level of *p* < 0.001.

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
