# Peer review of "Betanin, a Natural Food Additive: Stability, Bioavailability, Antioxidant and Preservative Ability Assessments"

_molecules, 2019, doi:10.3390/molecules24030458_

Round 1
Reviewer 1 Report
The study is interesting and useful to the community.
few suggested changes to improve the MS .
L106-107 the peak identified in Fig 1C as 1’ is not corresponding that shown in fig 1A and 1B. It looks like the 3rd unidentified peak in fig 1A. Please fix
L106 “4% of the original betanin mass“ please note that the concentration of the solutes in the beetroot juice is not 100% betanin, therefore the statement is not correct. Please fix.
L118-120 the argument is not relevant to the purified compound in fig 2B where there is no 637+ m/z ion
L144-146 “Furthermore, betanin would also display the ability to degrade and regenerate continuously during storage, thus maintaining betanin concentrations [22].” Please extend the discussion to explain how regeneration of betanin could be possible during storage?
L182-185 “The effect of the addition of purified betanin on other conservation parameters, such as texture, aroma, flavor, color and protein oxidation can be assessed not only in meats but also in other food products maintained under refrigeration.” This sentence does not make sense since the authors did not measure any of the properties mentioned!!
L248-250 “The antioxidant activities of purified betanin, following the sequential digestive process through the gastrointestinal apparatus, were increased when compared to pre-digested samples, as demonstrated in the FRAP, TEAC and ORAC assays, as well as the total antioxidant potential…” This statement is not correct as there is not statistical difference between pre-digestion and end of digestion. Why the antioxidant activities increase in the small intestines? Is it the effect of betanin or just the pH of the assay/sample?? Please discuss.
L253 “discretely increased in the post-oral digestion….” If it is not significantly different then there is no increase.
Author Response
Manuscript ID: Molecules-434942
Title: Betanin, a natural food additive: stability, bioavailability, antioxidant and preservative ability assessments
Authors: Davi Vieira Teixeira da Silva, Diego dos Santos Baião, Fabrício de Oliveira Silva, Genilton Alves, Daniel Perrone, Eduardo Mere Del Aguila and Vania M. Flosi Paschoalin*
General Comments by the authors:
We believe that we have fully addressed and understood Reviewers 1, 2 and 3 concerns and comments.
Four new references were included:
18. Herbach, K. M.; Stintzing, F. C.; Carle, R. Betalain stability and degradation-structural and chromatic aspects. J. Food Sci. 2006, 71, R41–R50, doi: https://doi.org/10.1111/j.1750-3841.2006.00022.x.
19. Cai, Y.; Sun, M.; Corke, H. Antioxidant activity of betalains from plants of the Amaranthaceae. J. Agric. Food. Chem. 2003, 51, 2288–2294, doi: 10.1021/jf030045u.
24. Kujala, T.S.; Loponen, J.M.; Klika, K.D.; Pihlaja, K. Phenolics and betacyanins in red beetroot (Beta vulgaris) root: distribution and effect of cold storage on the content of total phenolics and three individual compounds. J. Agric. Food Chem. 2000, 48, 5338-5342, doi: 10.1021/jf000523q
36. Frank, T.; Stintzing, F. C.; Carle, R.; Bitsch, I.; Quaas, D.; Strass, G.; Bitsch, R.; Netzel, M. Urinary pharmacokinetics of betalains following consumption of red beet juice in healthy humans. Pharmacol. Res. 2005, 52, 290–297, doi: 10.1016/j.phrs.2005.04.005.
Figure 1 was modified and the chemical structure of betanin was included.
The abstract and text were rephrased according to the reviewer's suggestions, following the order of the topic results. The entire text was revised to improve understanding (as requested by reviewer 2).
All modifications were highlighted in yellow.
After modifications, the manuscript was revised by a specialized editing company in order to improve English grammar and syntax.
The modifications have increased the overall impact of the manuscript. We would like to thank the reviewers for their insights and thoughtful critique of our manuscript.
Reviewer comments precede our responses.
Answers to Reviewer 1
L106-107 the peak identified in Fig 1C as 1’ is not corresponding that shown in fig 1A and 1B. It looks like the 3rd unidentified peak in fig 1A. Please fix.
Answer: The peaks identified in Figure 1A are commercial standard betanin (peak 1) and isobetanin (peak 1'). The isobetanin symbol in Figure 1A was corrected (peak 1'). In figure 1B, we identified betanin and isobetanin in beetroot through semi-preparative HPLC. However, Figure 1C displays the purified betanin obtained through semi-preparative HPLC followed by separation using an analytical HPLC column. The analytical column used for the separation and the elution conditions in Figure 1C was not able to clearly separate the peaks at the initial isomerization stage. At the beginning of the isomerization of betanin to isobetanin, a co-elution and a slight enlargement of the base occurred. This condition was included and discussed in the text (page 3, line 107).
L106 “4% of the original betanin mass“ please note that the concentration of the solutes in the beetroot juice is not 100% betanin, therefore the statement is not correct. Please fix.
Answer: The sentence on page 3, line 103 was rewritten according to the reviewer's correction. These 4% (48 mg.mL-1) correspond to the concentration of betanin recovered from beetroot juice (1.19 g.mL-1).
L118-120 the argument is not relevant to the purified compound in fig 2B where there is no 637+ m/z ion.
Answer: The sentence on page 4, line 130 was rewritten according to the reviewer's correction.
The 637+ m/z compound is not an ion. It is an adduct formed by condensation of the formic acid (molecular weight 46) used in the purification with decarloxylated betanin derivative, forming 6'-O-malonyl-2-descarboxyl-betanin (molecular weight 592), a decarboxylated betanin derivative [21].
Reference:
[21] Belhadj Slimen, I.; Najar, T.; Abderrabba, M. Chemical and antioxidant properties of betalains. J. Agric. Food Chem. 2017, 65, 675–689, doi: 10.1021/acs.jafc.6b04208.
L144-146 “Furthermore, betanin would also display the ability to degrade and regenerate continuously during storage, thus maintaining betanin concentrations [22].” Please extend the discussion to explain how regeneration of betanin could be possible during storage?
Answer: The sentence on page 5, line 157 was rewritten according to the reviewer's correction.
Factors such as temperature, pH, and type of buffer solution and the presence or absence of oxygen affect betanin stability during storage. However, betanin can also display the ability to degrade and regenerate continuously during storage, thus maintaining betanin concentrations [23]. This continuous betanin regeneration capacity during the storage process is still not well elucidated in the literature [2,23,24].
References:
2. Azeredo, M.C. Betalains: properties, sources, applications, and stability – a review. Int. J. Food Sci. Technol. 2009, 44, 2365–2376, doi: https://doi.org/10.1111/j.1365-2621.2007.01668.x.
23. Han, D.; Kim, S.J.; Kim, S.H.; Kim, D.M. Repeated regeneration of degraded red beet juice pigments in the presence of antioxidants. J. Food Sci. 1998, 63, 69–72, doi: https://doi.org/10.1111/j.1365-2621.1998.tb15678.x.
24. Kujala, T.S.; Loponen, J.M.; Klika, K.D.; Pihlaja, K. Phenolics and betacyanins in red beetroot (Beta vulgaris) root: distribution and effect of cold storage on the content of total phenolics and three individual compounds. J. Agric. Food Chem. 2000, 48, 5338-5342, doi: 10.1021/jf000523q.
L182-185 “The effect of the addition of purified betanin on other conservation parameters, such as texture, aroma, flavor, color and protein oxidation can be assessed not only in meats but also in other food products maintained under refrigeration.” This sentence does not make sense since the authors did not measure any of the properties mentioned!!
Answer: The reviewer is correct. The sentence on page 6, line 208 was removed according to the reviewer's correction.
L248-250 “The antioxidant activities of purified betanin, following the sequential digestive process through the gastrointestinal apparatus, were increased when compared to pre-digested samples, as demonstrated in the FRAP, TEAC and ORAC assays, as well as the total antioxidant potential…” This statement is not correct as there is not statistical difference between pre-digestion and end of digestion. Why the antioxidant activities increase in the small intestines? Is it the effect of betanin or just the pH of the assay/sample?? Please discuss.
Answer: The sentence on page 8, line 263 was rewritten according to the reviewer's correction.
As explained in the manuscript, page 8, line 272-308, the pH of the different fluids in the human body undergoes variations, influencing the stability and bioactivity of betanin in the different digestive tract compartments. The antioxidant ability of betanin was reduced in the acidic pH of the gastric fluid, evidencing the pH-dependence of the free radical-scavenging activity of betanin, but countering its antioxidant activity in the alkaline environment of the small intestine lumen, where the antioxidant activity assessed by all assays was either increased or maintained at the same level as the oral fluid, as mentioned previously. The alternate antioxidant activity of betanin between low and high levels in the simulated gastric compartment and in the small intestine indicates that the chemical structure of betanin was unaltered following acidic pH exposure during simulated gastric digestion, but should be attributed to the protonation of the betanin molecule, favoring the maintenance of free radical-scavenging activity until absorption in the gastrointestinal tract.
Furthermore, changes in pH fluids induce changes in the chemical structure of betanin along the gastroenteric apparatus [35]. The increase in betanin bioactivity when in an alkaline pH environment as found in the simulated small intestine fluid can be ascribed to its ability to donate H+ and electrons when altering from the cationic to the mono, bis and tri deprotonated states. The free radical scavenging activity of betanin at different pH (from 2 to 9) was previously determined through the TEAC assay, phenolic O-H homolytic bond dissociation energy (OH BDE), ionization potential (IP) and deprotonation energy (DE) [36]. This implies that, at slightly alkaline pH, betanin becomes a better hydrogen and electron donor, increasing its radical-scavenger ability as observed in the antioxidant assays in the simulated small intestine fluid when compared to the gastric fluid.
Reference:
35. Frank, T.; Stintzing, F. C.; Carle, R.; Bitsch, I.; Quaas, D.; Strass, G.; Bitsch, R.; Netzel, M. Urinary pharmacokinetics of betalains following consumption of red beet juice in healthy humans. Pharmacol. Res. 2005, 52, 290–297, doi: 10.1016/j.phrs.2005.04.005.
36. Gliszczyńska-Swigło, A.; Szymusiak, H.; Malinowska, P. Betanin, the main pigment of red beet: molecular origin of its exceptionally high free radical-scavenging activity. Food Addit. Contam. 2006, 23, 1079–1087, doi: 10.1080/02652030600986032.
L253 “discretely increased in the post-oral digestion….” If it is not significantly different then there is no increase.
Answer: The sentence in page 8, line 267 was rewritten according to the reviewer's correction.

Reviewer 2 Report
Reviewed manuscript show very interesting and comprehensive data, which are clearly presented. I have only one suggestion to Authors. In my opinion chemical structure of betanin and isobetanin on figure 1 ought to be added.
Author Response
Manuscript ID: Molecules-434942
Title: Betanin, a natural food additive: stability, bioavailability, antioxidant and preservative ability assessments
Authors: Davi Vieira Teixeira da Silva, Diego dos Santos Baião, Fabrício de Oliveira Silva, Genilton Alves, Daniel Perrone, Eduardo Mere Del Aguila and Vania M. Flosi Paschoalin*
General Comments by the authors:
We believe that we have fully addressed and understood Reviewers 1, 2 and 3 concerns and comments.
Four new references were included:
18. Herbach, K. M.; Stintzing, F. C.; Carle, R. Betalain stability and degradation-structural and chromatic aspects. J. Food Sci. 2006, 71, R41–R50, doi: https://doi.org/10.1111/j.1750-3841.2006.00022.x.
19. Cai, Y.; Sun, M.; Corke, H. Antioxidant activity of betalains from plants of the Amaranthaceae. J. Agric. Food. Chem. 2003, 51, 2288–2294, doi: 10.1021/jf030045u.
24. Kujala, T.S.; Loponen, J.M.; Klika, K.D.; Pihlaja, K. Phenolics and betacyanins in red beetroot (Beta vulgaris) root: distribution and effect of cold storage on the content of total phenolics and three individual compounds. J. Agric. Food Chem. 2000, 48, 5338-5342, doi: 10.1021/jf000523q
36. Frank, T.; Stintzing, F. C.; Carle, R.; Bitsch, I.; Quaas, D.; Strass, G.; Bitsch, R.; Netzel, M. Urinary pharmacokinetics of betalains following consumption of red beet juice in healthy humans. Pharmacol. Res. 2005, 52, 290–297, doi: 10.1016/j.phrs.2005.04.005.
Figure 1 was modified and the chemical structure of betanin was included.
The abstract and text were rephrased according to the reviewer's suggestions, following the order of the topic results. The entire text was revised to improve understanding (as requested by reviewer 2).
All modifications were highlighted in yellow.
After modifications, the manuscript was revised by a specialized editing company in order to improve English grammar and syntax.
The modifications have increased the overall impact of the manuscript. We would like to thank the reviewers for their insights and thoughtful critique of our manuscript.
Reviewer comments precede our responses.
Answers to Reviewer 2
Some notes should be clear. For examples,
Reviewed manuscript show very interesting and comprehensive data, which are clearly presented. I have only one suggestion to Authors. In my opinion chemical structure of betanin and isobetanin on figure ought to be added.
Answer: Figure 1 was modified and the chemical structure of betanin was included (page 3). Isobetanin is an optical betanin isomer. Isobetanin has a different spatial conformation in the carbonyl group at carbon 15 when compared to betanin. Thus, only the molecular structure of betanin was included in Figure 1 (the figure was sourced from Cai et al., 2003).
Reference:
Cai, Y.; Sun, M.; Corke, H. Antioxidant activity of betalains from plants of the Amaranthaceae. J. Agric. Food. Chem. 2003, 51, 2288-2294, doi: 10.1021/jf030045u.

Reviewer 3 Report
The manuscript numebered "molecules-434942" is well-written and organized. The topic is interesting since it deals with the possibility to use a natural low-toxicity compound, betanin, as a food preservative. To this end, the authors studied its stability and antioxidant ability during a human digestion simulation. In my view, the manuscript is ready for publication. There are only minor points to revise:
- abstract and through the manuscript: please, delete the sign + for the ions; for example, write m/z 551.16 (m/z before the number) instead of m/z 551.16+
- Figure 1: actually, the panel C (semipreparative purification) shows a peak with a shoulder indicating the trace occurrence of isobetanin; it should be indicated in the text that the semipreparative column has not the same efficiency as the analytical column (panels A and B).
- line 121: please, rephrase. For example: "The mass spectrometry analysis allowed one to confirm that the purification procedure succeeded in isolating betanin".
- as an example, see lines 134 and 471: please, uniform through the manuscript the way to write measure units (sometimes you expressed them as a ratio and sometimes as a power).
- line 139: "....who assessed the product stability during...."
- line 154, MDA and TBARS: you should explain the acronym the first time they occur in the text.
- Tables 1 and 2: there is a problem with the significant digits. Usually, standard deviations can be expressed using from 1 up to a maximum of three digits (I'm talking about significant not decimal digits); then, if you establish to use the a specific number of digits (for example 1), this rule should be respected in all cases. This also means that the associated measurement should be corrected consequently; for example:
23.0 ± 0.6 (Table 1), 14.80 ± 0.07 (Table 1), 1000 ± 200 (Table 2), 4300 ± 700 (Table 2), etc.
- line 355: MS identification should be based on the exact mass not on the molecular weight.
- line 379: were
Author Response
Manuscript ID: Molecules-434942
Title: Betanin, a natural food additive: stability, bioavailability, antioxidant and preservative ability assessments
Authors: Davi Vieira Teixeira da Silva, Diego dos Santos Baião, Fabrício de Oliveira Silva, Genilton Alves, Daniel Perrone, Eduardo Mere Del Aguila and Vania M. Flosi Paschoalin*
General Comments by the authors:
We believe that we have fully addressed and understood Reviewers 1, 2 and 3 concerns and comments.
Four new references were included:
18. Herbach, K. M.; Stintzing, F. C.; Carle, R. Betalain stability and degradation-structural and chromatic aspects. J. Food Sci. 2006, 71, R41–R50, doi: https://doi.org/10.1111/j.1750-3841.2006.00022.x.
19. Cai, Y.; Sun, M.; Corke, H. Antioxidant activity of betalains from plants of the Amaranthaceae. J. Agric. Food. Chem. 2003, 51, 2288–2294, doi: 10.1021/jf030045u.
24. Kujala, T.S.; Loponen, J.M.; Klika, K.D.; Pihlaja, K. Phenolics and betacyanins in red beetroot (Beta vulgaris) root: distribution and effect of cold storage on the content of total phenolics and three individual compounds. J. Agric. Food Chem. 2000, 48, 5338-5342, doi: 10.1021/jf000523q
36. Frank, T.; Stintzing, F. C.; Carle, R.; Bitsch, I.; Quaas, D.; Strass, G.; Bitsch, R.; Netzel, M. Urinary pharmacokinetics of betalains following consumption of red beet juice in healthy humans. Pharmacol. Res. 2005, 52, 290–297, doi: 10.1016/j.phrs.2005.04.005.
Figure 1 was modified and the chemical structure of betanin was included.
The abstract and text were rephrased according to the reviewer's suggestions, following the order of the topic results. The entire text was revised to improve understanding (as requested by reviewer 2).
All modifications were highlighted in yellow.
After modifications, the manuscript was revised by a specialized editing company in order to improve English grammar and syntax.
The modifications have increased the overall impact of the manuscript. We would like to thank the reviewers for their insights and thoughtful critique of our manuscript.
Reviewer comments precede our responses.
Answers to Reviewer 3
The manuscript numbered “molecules-434942” is well-written and organized. The topic is interesting since it deals with the possibility to use a natural low-toxicity compound, betanin, as a food preservative. To this end, the authors studied its stability and antioxidant ability during a human digestion simulation. In my view, the manuscript is ready for publication. There are only minor points to revise:
-abstract and through the manuscript: please, delete the sign + for the ions; for example, write m/z 551.16 (m/z before the number) instead of m/z 551.16+
Answer: The sign + for the ions was deleted and “m/z” was placed before the molecular mass throughout the entire manuscript, as suggested by the reviewer.
- Figure 1: actually, the panel C (semipreparative purification) shows a peak with a shoulder indicating the trace occurance of isobetanin; it should be indicated in the text that the semipreparative column has not the same efficiency as the analytical column (panels A and B).
Answer: The sentence on page 3, line 107 was rewritten according to the reviewer's correction.
-line 121: please, rephrase. For example: “the mass spectrometry analysis allowed one to confirm that the purification procedure succeeded in isolating betanin”.
Answer: The sentence on page 4, line 133 was rewritten according to the reviewer's correction.
- As an example, see lines 134 and 471: please, uniform through the manuscript the way to write measure units (sometimes you expressed them as a ratio and sometimes as a power).
Answer: This was modified and standardized according to the reviewer's correction.
- line 139:“…. who assessed the product stability during….”
Answer: The sentence on page 4, line 151 was modified according to the reviewer's correction.
- Line 154, MDA and TBARS: you should explain the acronym the first time they occur in the text.
Answer: MDA and TBARS acronyms were explained the first time they were noted in the text (page 5, line 171 and 172).
Tables 1 and 2: there is a problem with the significant digits. Usually, standard deviations can be expressed using from 1 up to a maximum of three digits (I'm talking about significant not decimal digits); then, if you establish to use the a specific number of digits (for example 1), this rule should be respected in all cases. This also means that the associated measurement should be corrected consequently; for example:
23.0 ± 0.6 (Table 1), 14.80 ± 0.07 (Table 1), 1000 ± 200 (Table 2), 4300 ± 700 (Table 2), etc.
Answer: The significant digits in all tables were uniformed according to the reviewer's correction.
- line 355: MS identification should be based on the exact mass not on the molecular weight.
Answer: The sentence on page 10, line 366 was rewritten according to the reviewer's correction.
- line 379: were
Answer: The sentence on page 10, line 388 and 390 were rewritten according to the reviewer's correction.
